# Longitudinal Associations between the Neighborhood Built Environment and Cognition in US Older Adults: The Multi-Ethnic Study of Atherosclerosis

**DOI:** 10.3390/ijerph18157973

**Published:** 2021-07-28

**Authors:** Lilah M. Besser, Lun-Ching Chang, Jana A. Hirsch, Daniel A. Rodriguez, John Renne, Stephen R. Rapp, Annette L. Fitzpatrick, Susan R. Heckbert, Joel D. Kaufman, Timothy M. Hughes

**Affiliations:** 1Department of Urban and Regional Planning and Institute for Human Health and Disease Intervention, Florida Atlantic University, Boca Raton, FL 33431, USA; 2Department of Mathematical Sciences, Florida Atlantic University, Boca Raton, FL 33431, USA; changl@fau.edu; 3Urban Health Collaborative and Department of Epidemiology and Biostatistics, Dornsife School of Public Health, Drexel University, Philadelphia, PA 19104, USA; jah474@drexel.edu; 4Department of City and Regional Planning, Institute for Transportation Studies, University of California Berkeley, Berkeley, CA 94720, USA; danrod@berkeley.edu; 5Department of Urban and Regional Planning, Florida Atlantic University, Boca Raton, FL 33431, USA; jrenne@fau.edu; 6Department of Psychiatry & Behavioral Medicine, Wake Forest School of Medicine, Winston-Salem, NC 27109, USA; srapp@wakehealth.edu; 7Departments of Family Medicine, Epidemiology, and Global Health, University of Washington, Seattle, WA 98105, USA; fitzpal@uw.edu; 8Cardiovascular Health Research Unit, Department of Epidemiology, University of Washington, Seattle, WA 98105, USA; heckbert@uw.edu; 9Departments of Environmental and Occupational Health Sciences, Medicine, and Epidemiology, University of Washington, Seattle, WA 98105, USA; joelk@uw.edu; 10Department of Internal Medicine, Department of Epidemiology and Prevention, Wake Forest School of Medicine, Winston-Salem, NC 27109, USA; tmhughes@wakehealth.edu

**Keywords:** built environment, residence characteristics, city planning, cognition, Alzheimer disease, preventive medicine, public health, walking destination, land use, older adult

## Abstract

Few studies have examined associations between neighborhood built environments (BE) and longitudinally measured cognition. We examined whether four BE characteristics were associated with six-year change in global cognition and processing speed. We obtained data on 1816 participants without dementia from the Multi-Ethnic Study of Atherosclerosis. BE measures included social destination density, walking destination density, proportion of land dedicated to retail, and network ratio (street connectivity). Global cognition was measured with the Cognitive Abilities Screening Instrument (CASI) and processing speed with the Digit Symbol Coding test (DSC). Multivariable random intercept logistic models tested associations between neighborhood BE at 2010–2012 and maintained/improved cognition (versus decline) from 2010–2018, and mediation by minutes of physical activity (PA)/week. The sample was an average of 67 years old (standard deviation = 8.2) (first cognitive measurement) and racially/ethnically diverse (29% African American, 11% Chinese, 17% Hispanic, 44% White). Compared to individuals with no walking destinations in the 1-mile surrounding their residence, those with 716 walking destinations (maximum observed) were 1.24 times more likely to have maintain/improved DSC score (Odds ratio: 1.24; 95% confidence interval: 1.03–1.45). No other associations were observed between BE and cognition, and PA minutes/week did not mediate the association between walking destination density and DSC change. This study provides limited evidence for an association between greater neighborhood walking destinations and maintained/improved processing speed in older age and no evidence for associations between the other BE characteristics and cognition. Future studies with finer grained BE and cognitive measures and longer-term follow up may be required.

## 1. Introduction

Approximately 15% of older adults (≥65 years) experience mild cognitive impairment [1] characterized by a decline in cognitive functioning that is significantly worse than previously attained abilities. Cognitive tests are used by clinicians and researchers to assess cognitive functioning and potential decline in the domains (e.g., memory, language, attention, executive function, global cognition) commonly affected in neurodegenerative diseases such as Alzheimer’s disease. Cognitive decline can also be a part of normal aging. Age and genetics/family history are strong risk factors for developing cognitive decline [2]. However, these factors are not modifiable. With the expected rise in individuals experiencing cognitive decline, paralleling the growing population of older adults [3], it is increasingly important to determine modifiable environmental factors that help prevent/delay the onset of cognitive decline.

Neighborhood built environments (BE), including streets, sidewalks, buildings, parks, and other human-made spaces/infrastructure, have been associated with health behaviors such as physical activity (PA) and health outcomes such as depression [4,5,6,7]. Preliminary evidence also suggests neighborhood BE characteristics are associated with late-life cognition and dementia, which is characterized by a significant decline in cognition to the point of affecting activities of daily living (e.g., preparing meals). For example, neighborhoods with mixed land uses (e.g., residential and retail) have been associated with lower odds of dementia [8] and greater neighborhood park space in early- and mid-life has been associated with less late-life decline on a cognitive measure of intelligence [9]. Neighborhoods with access to public transit, community centers, and public spaces in good condition have been associated with slower late-life decline in global cognition [10]. Also, greater street connectivity (number of options/choices for travel routes) has been associated with slower decline in late-life attention [11].

One potential mechanism to explain these associations is that mixed land uses, green spaces, and greater street connectivity encourage neighborhood walking and thereby increased PA, which has been associated with better cognition and lower dementia risk [12,13]. In addition, BE-cognition associations may be due to possible effects of neighborhood environments on mental health, cognitive stimulation, social engagement, or air pollution exposure. The causal mechanisms were not explicated, but it seems likely that more than one of these putative mechanisms help explain BE-cognition associations, as each has been associated with cognition [14,15,16,17,18,19].

Studies examining BE, cognitive decline, and dementia risk in older adults are limited [20,21,22,23,24] and few studies examined longitudinal associations [9,10,11,24,25]. Yet, increasing attention has been directed at the importance of social determinants of health such as the BE over the life course for brain health, and thus more studies of the BE and cognition are needed [26]. Cross-sectional differences in cognitive functioning may be due to factors such as early-life socioeconomic status or cultural differences affecting testing and differential geographic distribution by these SES/cultural characteristics. Beyond establishing a temporal sequence of BE exposure and cognitive change, longitudinal study designs allow investigation of changes in cognition that may reflect underlying brain aging or disease processes and not simply baseline differences among individuals. Large-scale intervention studies on the BE are difficult to implement, and therefore, studies using longitudinal measures provide supportive evidence for causal associations between the BE and cognition. With the ultimate goal of determining BE characteristics that may be addressable via community-based interventions and policies, this study investigated four neighborhood BE measures representing land use mix, destination accessibility, and street connectivity hypothesized to be associated with cognitive change based on previous associations with late-life cognition [10,11,22,23]. We hypothesized that individuals living in neighborhoods with higher destination densities, increased street connectivity, and greater access to retail destinations will have increased opportunities for social, mental, and physical activity, and thus will demonstrate maintained/improved cognition over time. In addition, we hypothesized that PA partially mediates the BE-cognition associations.

## 2. Materials and Methods

### 2.1. Sample

We obtained longitudinal data from the Multi-Ethnic Study of Atherosclerosis (MESA), a population-based cohort study of subclinical cardiovascular disease. This study only used de-identified data (i.e., not human subjects research). MESA participants were enrolled in 2000–2002 from six US cities (Forsyth County, North Carolina; New York, New York; Baltimore, Maryland; St. Paul, Minnesota; Chicago, Illinois; Los Angeles, California) and completed six in-person exams to date. At each exam, data are collected on demographics, medical history and health status, medications, anthropometry, self-reported PA, as well as a number of procedures and assessments (e.g., blood pressure, imaging). Details on MESA are found elsewhere [27]. We restricted individuals to those with cognitive data at Exam 5 (2010–2012) and Exam 6 (2016–2018), no ICD-9 diagnoses for dementia determined from hospitalization or death records [28], no dementia medication use, ≥1 non-missing BE measures, <3 missing items on the Cognitive Abilities Screening Instrument (CASI), and CASI scores > 20 (scores < 20 lack face validity).

### 2.2. Measures

#### 2.2.1. Cognitive Test Scores

The validated [29] Cognitive Abilities Screening Instrument (CASI, version 2) [30] is a global measure of cognitive function assessing attention, concentration, orientation, abstraction, judgment, verbal fluency, language, short-term and long-term memory, and visual construction (range: 0–100). The validated [31] Digit Symbol Coding (DSC) task [32] measures processing speed (range: 0–133). Higher scores indicate better cognitive function. In prior cross-sectional MESA studies [21,33], the Digit Span test measuring working memory was less consistently associated with BE measures compared to the CASI and DSC and thus was excluded to reduce multiple comparisons.

Dichotomous measures of cognitive change were of primary interest because we aimed to understand if the BE measures were associated with maintained or improved cognition over time specifically. Simply the maintenance of cognition over time may be of clinical significance among older adults because they often experience cognitive decline due to normal aging. The dichotomous measures of longitudinal change in CASI and DSC scores were calculated from continuous measures of annual change in scores from Exam 5 to 6. We ran unadjusted linear regression for each participant with cognitive score as the outcome and time as the predictor (Exam 5 = 0, Exam 6 = years since Exam 5). Then, the dichotomous outcomes were derived by categorizing the resulting continuous measures as maintained/improved score (i.e., score change = 0 or increase in score from Exam 5 to 6) versus decline in score from Exam 5 to 6. Regression analyses focused on continuous cognitive outcomes were based on z-scores. Raw CASI and DSC scores were transformed into z-scores by subtracting from each participant’s score the mean score of the sample and dividing the result by the standard deviation of the sample.

#### 2.2.2. Neighborhood Built Environment Measures

Measures of social destination density, walking destination density, network ratio, and proportion of land dedicated to zoned retail (hereafter termed proportion retail), were previously developed in the MESA Neighborhood Study [34,35]. Participant residential addresses were geocoded for Exams 1 through 5 to derive BE measures for the circular/radial/Euclidian buffers surrounding participants’ homes. Any new residential addresses were accounted for when deriving BE measure values for each exam. Half-mile buffers were of primary interest since adults typically walk up to a half-mile to neighborhood destinations [36]. We also examined associations using 1-mile buffers, which were associated with health outcomes in prior MESA studies [37,38]. Full details have been previously published [33] on the types of social and walking destinations (source: National Establishment Time Series [39]) and retail establishments (source: county/municipal cadastral data [35]). Briefly, social destinations included places such as recreational clubs, beauty shops, and religious organizations, and walking destinations included places such as post offices, drug stores, and banks. The number of social and walking destinations was calculated per radial buffer to provide simple density measures. Retail establishments included destinations such as shopping centers and restaurants, and proportion retail was calculated as the proportion of the radial buffer composed of these retail land uses. Network ratio (source: Esri StreetMap), a measure of street connectivity, was calculated by dividing the buffer area measured by taking ½-mile distances from home using the street network (versus radial buffers) by the total radial buffer area [35]. Values of 1 indicate a very connected street network whereas values of zero indicates a totally disconnected network.

#### 2.2.3. Other Neighborhood Measures

We used a single measure of neighborhood socioeconomic status (SES) (by US Census tract) that was previously derived from a principal components analysis of seven US Census American Community Survey (2007–2011) variables: neighborhood median home value, percentage rental income, and median household income, as well as percentage of neighborhood residents with high school degrees, bachelor’s degrees, annual household incomes >$50,000, and managerial occupations. Neighborhood population density was calculated for the radial buffers surrounding the participants’ homes based on the 2010 Census block-level population density. Assuming an equal distribution of the population per block, the population/km^2^ was calculated for each radial buffer. Measures based on ½-mile radial buffers were used in this study. Appraisals of neighborhood safety walking and safety from crime were reported by participants on a 5-point scale (strongly agree to strongly disagree).

#### 2.2.4. Other Measures

Participant demographics included age, sex, race/ethnicity, education, car ownership, and income. Self-reported health status/behaviors from Exam 5 included depressive symptoms (Center for Epidemiological Studies Depression Scale [CES-D] ≥ 16), current smoker, diabetes mellitus, arthritis (any type), cardiovascular disease, cerebrovascular disease, and self-reported PA (minutes walking to places, total MET-minutes/week). Number of residential moves was available for the MESA follow-up period. Body mass index (kg/m^2^) calculations were based on height and weight measurements, and apolipoprotein E (APOE) genotype (genetic risk factor for Alzheimer’s disease) was dichotomized into APOE ε4 carriers (≥1 ε4 allele) and non-carriers.

### 2.3. Statistical Analyses

We calculated descriptive statistics (e.g., means, standard deviations) for the demographics, health status/behaviors, cognitive test scores, and neighborhood/BE characteristics. Unadjusted and adjusted random intercept logistic regression models (accounting for clustering by US Census tract) determined associations between the BE variables and CASI and DSC score changes from Exam 5 to Exam 6. Separate regression models were run for each BE-cognitive test combination. We controlled for covariates known a priori or hypothesized to be associated with BE and/or cognition: age (Exam 5), sex, education, race/ethnicity, income, neighborhood SES, site, APOE ε4 carriers, appraisal of neighborhood safety and crime, arthritis, cardiovascular and cerebrovascular disease, diabetes, and number of residential moves.

In sensitivity analyses, we re-ran the adjusted regression models described above while: (1) additionally controlling for car ownership, (2) replacing the dichotomous outcomes with continuous measures of cognitive change (z-scores) and employing adjusted random intercept linear regression (accounted for clustering by participant and US Census tract), and (3) replacing the dichotomous outcomes with measures of cognitive change based on three categories (>1 SD lower than mean change in score, within 1 SD of mean [reference group], >1 SD higher than mean). The latter sensitivity analysis was performed to account for potential learning effects in repeating the same test after six years, which might result in maintained/improved scores over time for some individuals. In the final sensitivity analysis, we calculated propensity scores for the probability of inclusion in our analytic sample based on age, education, sex, income, race/ethnicity, marital status, site, and comorbidities (measured at enrollment, Exam 1). The propensity scores were incorporated into inverse probability weights that were applied to the logistic regression models to address potential selection/attrition bias.

For BE-cognition associations with *p* < 0.05, the multivariable regression analyses were repeated in an exploratory analysis to assess mediation by total MET-minutes of PA/week using a causal steps approach [40]. We examined associations between the BE measure and cognitive score change, the BE measure and total PA/week, and total PA/week and cognitive score change. In addition, we explored whether change in the BE characteristics from Exam 1 to Exam 5 (including data from Exam 1, 2, 3, 4, and 5) was associated with dichotomous change in CASI and DSC scores from Exam 5 to Exam 6.

Regression analyses were conducted using the “lme4” package (“lm” function) in R version 4.0.2. (The R Foundation. Vienna, Austria.) No adjustments were made for multiple comparisons.

## 3. Results

The sample included 1816 participants (Figure 1). Mean age was 67.1 years at Exam 5 (standard deviation [SD] = 8.2), 53% were women, 75% had at least some college education, and 88% owned a car (Table 1). Thirty-six percent were obese (≥30 kg/m^2^), 13% had possible depression, 29% had arthritis, and 8% had cardiovascular and/or cerebrovascular disease. Twenty-nine percent reported frequently walking places (≥7 h/week). Seventy percent had no residential moves during MESA follow-up, 20% moved once, and 10% moved at least twice.

At Exam 5, mean CASI scores were 90.0 (SD = 6.8) and DSC scores were 55.4 (SD = 16.8) (Table 2). Maintained/increased CASI and DSC scores (Exam 5 to 6) were observed for 52% and 30%, respectively. Table 3 and Appendix A provide descriptive statistics for the neighborhood characteristics. For the majority of participants, annual changes in BE characteristics were small. Compared to non-movers, movers more often experienced a decrease in neighborhood social and walking destinations and street connectivity and an increase in proportion retail over time (Appendix A).

Appendix A provide unadjusted regression results. In adjusted analyses, none of the BE measures based on ½-mile buffers were associated with maintained/improved CASI or DSC scores (Table 4). Borderline associations (*p* < 0.10) were observed for greater walking destination density (per 100) (Odds ratio [OR]: 1.02; 95% confidence interval [CI]: 1.00, 1.03) and greater proportion retail (OR: 1.83; 95% CI: 0.98, 3.42) and maintained/improved DSC score. Using the 1-mile buffers, greater walking destination density (per 100) was associated with maintained/improved DSC score (OR: 1.03; 95% CI: 1.00, 1.05) (Table 4). Thus, compared to those with no walking destinations in the 1-mile surrounding their homes, those with 716 walking destinations (maximum observed in sample) were 1.24 times more likely to maintain/improve their DSC scores over 6 years (versus decline) (OR: 1.24; 95% CI: 1.03, 1.45). No other associations were observed between the BE characteristics and dichotomous cognitive outcomes.

In the first sensitivity analysis, additionally controlling for car ownership resulted in similar estimates to Table 4 (data not shown). In the adjusted regression analyses focused on continuous measures of change in CASI and DSC, no associations were found between the four BE measures and cognitive change (Appendix A). The BE measures were not associated with the three-category measures of longitudinal cognitive change (i.e., >1 SD lower than mean change, within 1 SD of mean [reference], >1 SD higher than mean) (data not shown), and use of inverse probability weights resulted in little change in the Table 4 estimates (Appendix A).

Total MET-minutes of PA/week was not a mediator of the association between walking destination density (1-mile buffer) and DSC score and was not associated with either measure (data not shown). Lastly, measures of average annual change in the BE characteristics from Exam 1 to 5 were not associated with the dichotomous measure of change in CASI or DSC scores from Exam 5 to 6 (Appendix A).

## 4. Discussion

Overall, this study provided little evidence for an association between the neighborhood BE and cognitive changes over time during late life (see Table 4). The main association observed was between neighborhood walking destination density and 6-year change in DSC scores. Compared to those with fewer neighborhood walking destinations, individuals living with higher neighborhood densities of walking destinations (1-mile around residence) had a greater odds of maintaining/improving processing speed over time.

We did not find associations for social destination density, network ratio, or proportion retail. Although preliminary, these results may suggest that destinations for conducting day-to-day transactions (e.g., bank) and shopping may encourage visiting neighborhood destinations and help to maintain late-life cognition. Despite finding that PA was not a mediator, other studies found that it was associated with processing speed and mediated BE-cognition associations [22,41]. Objective PA measures may be needed to detect mediation in future studies. Mediation by other factors including but not limited to air pollution, social engagement, and depression needs investigation to delineate the causal mechanism(s) relating the BE and cognition.

A cross-sectional study of older adults in Singapore found that land use mix and greater street connectivity were associated with better global cognition scores (Repeatable Battery for the Assessment of Neurocognitive Status) [22]. Another study, using cross-sectional MESA data, found that greater proportion retail was associated with better processing speed [21], proportion retail was not associated with global cognition (CASI) or working memory (Digit Span test), and measures of social and walking destination density and intersection density were inversely associated with global cognition and processing speed (CASI, DSC). Other cross-sectional studies found no or inverse associations between comparable BE measures and cognition [42,43]. While cross-sectional studies serve as preliminary evidence for associations, longitudinal studies are needed to help support assertions of causality.

Two longitudinal studies examined associations between similar BE measures and cognitive decline. A study of cognitively normal older adults in Kansas (US) found associations between greater neighborhood street connectivity and slower two-year decline in attention and between greater street integration (fewer turns to get to destinations) and greater two-year decline in attention and verbal memory [11]. Street connectivity measured the number of streets directly linked to every other street in the road network (weighted by distance from residence), while street integration measured the number of turns to get to all other street segments in the road network. In contrast, our measure of street connectivity measured accessibility of nearby streets based on a set distance traveled on the road network. Thus, differences in exposure and outcome measurement may explain their positive and inverse findings compared to our null findings.

The other longitudinal study of older adults in Chicago, Illinois (US), found presence of neighborhood community centers associated with slower decline in global cognition [10]. Presence of a community center, although a neighborhood destination, provides a measure of community resources and not an assessment of social destination density, limiting the comparison with our findings. Overall, the extant literature used varying BE and cognitive measures and different methods to define neighborhoods (e.g., self-report versus objective, radial buffers versus Census boundaries), which may account for inconsistent findings.

We found that annual changes in BE exposures in the ten years preceding the cognitive assessments were not associated with longitudinal cognitive changes. This is possibly explained by the relative invariance of the BE measures over time. Larger changes over time would generally be expected for individuals moving to dramatically different neighborhoods (e.g., urban to rural). Over the MESA follow-up period, 70% of the sample did not move residences, and among the movers, the majority did not experience dramatic changes in their BEs following moves. This study was not designed to focus on the effects of residential moves on cognition, but instead on how differences and changes in BE exposures may affect cognition after controlling for the number of residential moves. However, future studies of individuals who move to dramatically different BEs, compared to those who do not move, may provide fruitful insights into BE-cognition associations. More refined measures of BE changes or more complex modeling of BE-cognition relationships may be warranted in future studies, particularly those including life course BE exposures. Comparisons of BE typologies (e.g., urban neighborhoods with and without many social destinations) may help elucidate causal mechanisms and increase specificity of BE exposures associated with maintained/improved cognition. In addition, measures such as proportion retail will need refinement to include quantification of density and/or quality of retail establishments. Prior findings of moderate to high correlation between the BE measures and population density [21] precluded its inclusion in adjusted analyses, although new studies stratifying associations by urbanicity may be useful. Altogether, additional studies are needed to investigate whether cognition is associated with accumulated BE exposures and longitudinal BE changes, and whether these associations depend on the locale (e.g., urban, rural), specific BE typology, and number/types of residential moves.

Study strengths include a racially/ethnically-diverse, population-based cohort enrolled from six US cities. We examined longitudinal change in exposures and outcomes. The availability of extensive phenotyping in MESA allowed for consideration of PA as a mediator and important confounders such as neighborhood SES. In addition, we tested associations with ½-mile and 1-mile neighborhood measures, to evaluate if associations remained consistent across varying neighborhood boundaries.

Study limitations include lack of early- and mid-life BE measures and early-life SES measures beyond educational attainment, factors that may demonstrate stronger associations with cognition than late-life measures. Data were not available on other cognitive domains that may be associated with BE exposures, such as attention and visuospatial function. The self-reported PA measures may not adequately reflect actual PA, and objective measures obtained through devices such as accelerometers may result in different mediation findings. The association between walking destination density and processing speed may have resulted from multiple testing, and the observed association only applied to the dichotomous processing speed measure, not the continuous or three-category measures. Thus, the study findings are tentative and need replication in other cohorts. Time spent in the neighborhood environment was not available, and we were not able to account for self-selection into neighborhoods that may affect associations, such that individuals with lower risk for cognitive decline move to neighborhoods with more walking/retail destinations. Additionally, we did not capture BE exposure outside of the neighborhood, therefore, the BE measures do not represent overall BE exposures. Lastly, study findings may not be generalizable to other US regions or groups.

## 5. Conclusions

In this study, we investigated associations between neighborhood measures of land use mix and destination accessibility and late-life changes in global cognition and processing speed. Greater walking destination density was associated with maintained/improved processing speed over time, but no other BE measures were associated with cognitive change. Based on our findings and the extant literature, evidence is not yet conclusive to support community interventions for increased land use mix/destination accessibility to help maintain cognitive health in older age. Future studies with finer grained BE and cognitive measures and longer-term follow up may be required.

## Figures and Tables

**Figure 1 ijerph-18-07973-f001:**
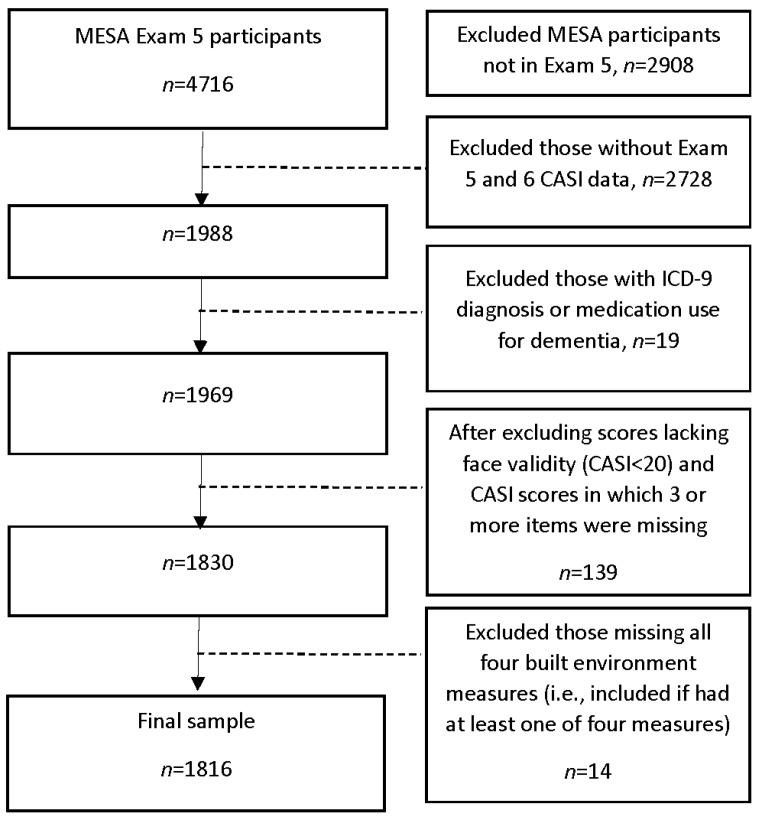
Sample size flow diagram. Abbreviations: MESA = Multi-Ethnic Study of Atherosclerosis; CASI = Cognitive Abilities Screening Instrument; ICD = International Classification of Diseases.

**Table 1 ijerph-18-07973-t001:** Participant characteristics.

Characteristic ^a^	*n* = 1816
Age at Exam 5, mean (SD)	
<60 years	372 (20.5%)
60–64 years	430 (23.7%)
65–69 years	321 (17.7%)
70–74 years	283 (15.6%)
75–79 years	266 (14.7%)
80 years or older	144 (7.9%)
Female, *n* (%)	957 (52.7%)
Education, *n* (%)	
<High school degree	165 (9.1%)
High school degree	280 (15.5%)
Some college, no bachelor’s degree	550 (30.4%)
Bachelor’s degree or higher	816 (45.1%)
Married, *n* (%)	1172 (65.2%)
Race/ethnicity, *n* (%)	
White/Caucasian	795 (43.8%)
Chinese-American	192 (10.6%)
Black/African American	521 (28.7%)
Hispanic	308 (17.0%)
Family income ≥ $30,000/year, *n* (%)	1392 (74.9%)
Own car, *n* (%)	1578 (87.8%)
≥1 APOE ε4 allele, *n* (%)	466 (27.1%)
Depressive symptoms (CES-D score ≥ 16), *n* (%)	224 (12.5%)
Current smoker, *n* (%)	114 (6.3%)
Obese (body mass index ≥ 30 kg/m^2^), *n* (%)	650 (35.8%)
Diabetes (self-reported), *n* (%)	164 (9.09)
Hypertension (taking medication), *n* (%)	905 (49.8%)
Taking depression medication, *n* (%)	229 (12.6%)
Arthritis (self-reported), *n* (%)	526 (29.3%)
Cardiovascular disease, *n* (%)	104 (5.7%)
Cerebrovascular disease (stroke/TIA), *n* (%)	38 (2.1%)
Frequently walk places (≥7 h/week), *n* (%)	529 (29.2%)
MET-minutes moderate/vigorous physical activity/week, mean (SD)	5783.6 (6564.3)
Number of moves, Exam 1 to Exam 5, mean (SD)	0.47 (0.89)
None	1269 (69.9%)
One	369 (20.3%)
Two or more	178 (9.8%)
Years between Exam 5 and Exam 6	6.3 (0.5)

Abbreviations: SD = standard deviation; APOE = apolipoprotein E; CES-D = Center for Epidemiologic Studies Depression Scale; TIA = transient ischemic attack; MET = metabolic equivalent of task; ^a^ Missing data: education, *n* = 5; married, *n* = 18; income, *n* = 42; APOE, *n* = 95; depression, *n* = 17; smoking, n = 19; obesity; *n* = 1; diabetes, *n* = 12; arthritis, *n* = 1; cerebrovascular disease, *n* = 1; cardiovascular disease, *n* = 1; walking places/physical activity, *n* = 5; own car, *n* = 18.

**Table 2 ijerph-18-07973-t002:** Cognitive test scores.

Score Measure ^a^	
Exam 5 cognitive test score, mean (SD)	
CASI (possible range: 0–100)	90.0 (6.8)
DSC (possible range: 0–133)	55.4 (16.8)
Change in cognitive test score, Exam 5 to 6, mean (SD)	
CASI	0.027 (1.012)
DSC	−0.728 (1.823)
Categorical change in CASI score, Exam 5 to 6, *n* (%)	
No change or increase in score	947 (52.2%)
Decrease in score	869 (47.9%)
Categorical change in DSC score, Exam 5 to 6, *n* (%)	
No change or increase in score	551 (30.3%)
Decrease in score	1265 (69.7%)

Abbreviations: SD = standard deviation; CASI = Cognitive Abilities Screening Instrument; DSC = Digit Symbol Coding; ^a^ Missing data: CASI at Exam 5, *n* = 0; DSC at Exam 5, *n* = 183; Change in CASI, Exam 5 to 6, *n* = 0; Change in DSC, Exam 5 to 6, *n* = 287.

**Table 3 ijerph-18-07973-t003:** Neighborhood characteristics.

Neighborhood Characteristic ^a^	Mean (SD)	Range: Lowest, Highest
Measured at Exam 5		
Social destination density ^b^	132.5 (217.2)	0, 1604.3
Walking destination density ^b^	60.0 (101.4)	0, 716.3
Network ratio ^b^	0.426 (0.183)	0.043, 0.801
Proportion retail ^b^	0.04 (0.05)	0.00, 0.30
Neighborhood socioeconomic status ^c,d^	−0.52 (1.20)	−4.22, 2.50
Population density ^a^	6462 (9474)	11, 54483
Average annual change, Exam 1 to 5		
Social destination density ^a^	2.8 (10.4)	−121.7, 111.7
Walking destination density ^a^	−0.5 (4.8)	−83.0, 49.5
Network ratio ^a^	−0.001 (0.013)	−0.069, 0.061
Proportion retail ^a^	−0.002 (0.007)	−0.067, 0.0312

Abbreviation: SD = Standard deviation. ^a^ Missing data: proportion retail, *n* = 118; neighborhood socioeconomic status, *n* = 32; ^b^ Measured in ½−mile radial buffer surrounding residence; ^c^ Measured at US Census tract level; ^d^ higher (more positive) value = worse SES.

**Table 4 ijerph-18-07973-t004:** Adjusted association between continuous built environment measures and dichotomous cognitive change measure.

At Exam 5 ^a^		Maintained/Improved CASI Score ^b,c^		Maintained/Improved DSC Score ^b,c^	
	Buffer Size	OR (95% CI)	*p*-Value	OR (95% CI)	*p*-Value
Social destination density (per 100)	½-mile	1.00 (0.99, 1.01)	0.98	1.00 (1.00, 1.01)	0.13
Walking destination density (per 100)	½-mile	1.00 (0.98, 1.02)	0.81	1.02 (1.00, 1.03)	0.07
Network ratio	½-mile	0.99 (0.83,1.17)	0.89	0.95 (0.81,1.10)	0.47
Proportion retail	½-mile	1.19 (0.59,2.40)	0.63	1.83 (0.98, 3.42)	0.06
Social destination density (per 100)	1-mile	1.00 (0.99, 1.01)	0.96	1.01 (1.00, 1.02)	0.09
Walking destination density (per 100)	1-mile	1.00 (1.00, 1.03)	0.83	**1.03 (1.00, 1.05)**	**0.02**
Network ratio	1-mile	1.05 (0.86, 1.26)	0.64	0.95 (0.81, 1.13)	0.56
Proportion retail	1-mile	1.57 (0.55, 4.43)	0.40	1.74 (0.69, 4.36)	0.24

Abbreviation: CI = Confidence Interval; CASI = Cognitive Abilities Screening Instrument; DSC = Digit Symbol Coding. ^a^ Continuous measures; ^b^ Dichotomized as maintained/improved score versus decline in score from Exam 5 to 6; ^c^ Controlling for age at Exam 5, sex, education, race/ethnicity, income, neighborhood socioeconomic status, site, APOE ε4 carrier, neighborhood perception of safety walking day or night and crime, arthritis, cardiovascular and cerebrovascular disease, diabetes, number of residential moves. Bold = *p* < 0.05.

## Data Availability

Data were obtained from the Multi-Ethnic Study of Atherosclerosis Coordinating Center. Data requests can be made to: https://mesa-nhlbi.org/ (accessed on 24 April 2021).

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
