# Peer review of "Longitudinal Associations between the Neighborhood Built Environment and Cognition in US Older Adults: The Multi-Ethnic Study of Atherosclerosis"

_ijerph, 2021, doi:10.3390/ijerph18157973_

Round 1
Reviewer 1 Report
Overall, this is an interesting topic that potentially adds to the existing knowledge of health in the built environment. However, some flaws affect the quality of the paper. In the following paragraphs, I discuss the strengths and weaknesses of this paper.
Abstract:
The abstract is well written and informative and provides all the information required. However, it is too long (297 words).
Introduction:
Overall, the introduction is well written and the importance of neighborhood BE on cognition. The transition from the introductory part to the research hypothesis is reasonable. However, the research gap is not well discussed. I would suggest adding one paragraph on how the lack of research on the effects of NBE on cognition drives the research purpose.
Methodology:
Using the input-output model for this research seems reasonable and I believe the authors have used the right way to apply this model in the research setting.
The sample and measures are well described and the recruitment strategy is logical.
Results:
The results of this research are interesting and provide some of the information necessary to move to the discussion section. However, some results are not reported properly such as PCA results with detailed tables and the procedure to select one component (neighborhood socioeconomic status) among all.
Furthermore, I would recommend reporting the fit indices of the analysis in order to provide support for the discussion.
Discussion:
Line 277: “Provided a little evidence…” may be supported by fit indices.
This section is well written and provides a good understanding of the findings. However, it lacks statistical support from the results section.
Conclusion:
The findings of the studies have been discussed in this section. However, this section is too short and does not include research limitations. Usually, the conclusion starts with the research purpose followed by a brief report of findings and limitations of the study and future work. It also is necessary to discuss the importance and the application of this research (broader impact) in this section.
Reviewer 2 Report
Thank you for the opportunity given to review this important paper. Congratulations to the author for their presentation. This is a good paper and deserved to be published - my opinion.
The only concern I have is the report on the surveys used in the study and that all the validity and reliability of the survey used should be reported in the methodology section.
